# Infrared-Inertial Navigation for Commercial Aircraft Precision Landing in Low Visibility and GPS-Denied Environments

**DOI:** 10.3390/s19020408

**Published:** 2019-01-20

**Authors:** Lei Zhang, Zhengjun Zhai, Lang He, Pengcheng Wen, Wensheng Niu

**Affiliations:** 1School of Computer Science and Engineering, Northwestern Polytechnical University, Dongda Road, Changan District, Xi’an 710072, China; zhaizjun@nwpu.edu.cn (Z.Z.); langhe@mail.nwpu.edu.cn (L.H.); 2Xi’an Aeronautics Computing Technique Research Institute, Aviation Industry Corporation of China, Jinyeer Road, Yanta District, Xi’an 710065, China; wpcheng@avic.com (P.W.); nwsheng@avic.com (W.N.)

**Keywords:** infrared-inertial navigation, homography, runway detection, observability analysis, precise landing, low visibility, GPS-denied

## Abstract

This paper proposes a novel infrared-inertial navigation method for the precise landing of commercial aircraft in low visibility and Global Position System (GPS)-denied environments. Within a Square-root Unscented Kalman Filter (SR_UKF), inertial measurement unit (IMU) data, forward-looking infrared (FLIR) images and airport geo-information are integrated to estimate the position, velocity and attitude of the aircraft during landing. Homography between the synthetic image and the real image which implicates the camera pose deviations is created as vision measurement. To accurately extract real runway features, the current results of runway detection are used as the prior knowledge for the next frame detection. To avoid possible homography decomposition solutions, it is directly converted to a vector and fed to the SR_UKF. Moreover, the proposed navigation system is proven to be observable by nonlinear observability analysis. Last but not least, a general aircraft was elaborately equipped with vision and inertial sensors to collect flight data for algorithm verification. The experimental results have demonstrated that the proposed method could be used for the precise landing of commercial aircraft in low visibility and GPS-denied environments.

## 1. Introduction

Landing is the most accident-prone phase of flight for both military and civil aircraft. This is due to the manoeuvring sequence required to exhaust a large amount of aircraft kinetic energy in a relatively small area. Fixed-wing aircraft usually descend smoothly at a constant angle, pointing in the direction of the runway centerline, and touch down at the beginning of the runway. If low visibility conditions (e.g., fog or haze) are encountered, the pilots have no choice but to manipulate the aircraft to land using navigation instruments. If the conventional radio navigation systems are disturbed or disabled, they can mislead the pilots and cause a Controlled Flight Into Terrain (CFIT) accident. In addition, most airports are equipped with simple and coarse radio beacons rather than expensive and precise ground-based guidance systems. Nowadays neither avionics systems, nor airport infrastructures are perfectly designed to support precision landing. Faced with these challenges, an autonomous, accurate and affordable landing navigation mechanism is extremely necessary for most fixed-wing aircrafts.

The traditional landing aid systems include the Instrument Landing System (ILS), and Global Position System (GPS). However, these systems themselves have deficiencies for fixed-wing aircraft precision landing. ILS can only guide an aircraft to the decision height (DH, usually DH = 100 ft) and cannot guide it onto the runway. Besides, ILS, with its high cost and complicated maintenance is not suitable for general aviation airports. Although GPS can meet the needs of class I and II landings for most aircraft, its signal is vulnerable to jamming or disabling [1]. 

Recently vision-based landing navigation, which has the benefits of accuracy, autonomy and low cost, is becoming a central research topic [2,3]. Existing studies on vision-based landing for fixed-wing aircraft are classified into two categories, namely ground-based and onboard-based. The ground-based methods [4,5,6,7,8,9,10] often utilize multilocular vision systems arranged on the ground to detect, track, and guide the aircraft to landing. Martínez et al. [4] designed a trinocular system, which is composed of three Firewire cameras fixed on the ground, to estimate an UAV’s position and orientation by tracking color land markers on the UAV. Kong et al. [5,6,7] developed a custom-built infrared stereo camera with a large field of view and claimed that their system could resist all weather conditions, and further improve the detection precision by the Chan-Vese method [8] and the saliency-inspired method [9]. In addition, Yang et al. [10] showed promising results for UAV auto-landing in GPS-denied environments using a ground-based infrared camera array and a near infrared laser lamp-based cooperative optical imaging method.

Onboard-based vision landing navigation based on looking-forward images and computer vision algorithms can be divided into two types, namely moving platform-based and airport runway-based methods. For landing on a moving platform, the core solution is to track a known target, e.g., an aircraft carrier, and compute its relative position and orientation. Coutard et al. at the French INRIA [11,12] proposed a method for carrier visual detection and tracking for landing on the deck. The carrier is detected in the image by a warped patch of a reference image. Ding et al. [13] presented a FLIR/INS/RA integrated landing guidance approach to estimate the aircraft states and carrier dynamics for fixed-wing aircraft landing on the deck in low-visibility weather and high sea states by employing the Newton iterative algorithm, Kalman filter and wavelet transform. Jia et al. [14] put forward a carrier landing algorithm based on point and line features for fixed-wing UAVs. This algorithm calculates the attitude according to the sky-sea line and runway vanishing point, estimates the position parameters on the basis of the landmark and tracklines’ collinear equations by least square solutions, but it was only verified by simulation experiments. Recently Muskardin et al. in DLR [15] analyzed and proposed an algorithm for a solar-powered fixed-wing UAV landing on top of a mobile ground vehicle. For landing on an airport runway, a fixed-wing aircraft should descend smoothly at a constant angle, pointed in the direction of the runway centerline, and touch down at the beginning of the runway. Korn et al. at DLR [16] proposed a simple method to estimate the relative of position of an aircraft with respect to a runway based only on camera images. Neither a calibrated camera, nor any knowledge of special points of the runway are needed. The premise of this method is to accurately detect the horizon, but it is not suitable for all airports. Goncalves et al. [17] presented a study of a vision-based automatic approach and landing for an aircraft use an efficient second-order minimization-based tracking method. In contrast with feature extraction methods, direct methods can achieve ideal accuracy, but they are computationally consuming. Gui et al. [18] proposed an airborne vision-based navigation approach for UAV accuracy landing based on artificial markers. This method needs to install a visible light camera integrated with a DSP processor on the UAV and place four infrared lamps on the runway. Guo et al. [19] designed a vision-aided landing navigation system based on a fixed waveband guidance illuminant using a single camera. Bras et al. [20] used the edges and the front corner points of the runway extracted from the forward-looking images to implement a visual servo control method for autonomous UAV landings. Fan et al. [21] adopted a spectral residual saliency map to detect regions of interest, then selected sparse coding and spatial pyramid matching to recognize runways and used orthogonal iteration to estimate position and attitude. Burlion et. al. at the French ONERA [22] studied the vision-based flight control problem under field of view constraints and proposed a vision-based landing framework for a fixed-wing UAV on an unknown runway. Gibert et al. at Airbus [23,24,25] designed two nonlinear observers based on a high gain approach and sliding mode theory and applied them to a vision-based solution for civil aircraft landing on an unknown runway. However, this method does not utilize inertial measurements with high update. Ruchanurucks et al. [26] used an Efficient Perspective-n-Point (EPnP) solution to estimate relative pose for an automatic aided landing system for landing a fixed-wing UAV on a runway. The accuracy of this method is however susceptible to runway detection errors.

Although the above algorithms have achieved remarkable progress in vision-aided landing navigation, there are four main problems which need to be coped with. Firstly, image sensors cannot satisfy the requirements of high-speed landing navigation because of their low image update rates, whereas, IMUs can measure accelerations and rotational velocities at high update rates. The two types of sensors can complement each other in nature and integrate well in an optimized framework [27]. Secondly, most often fixed-wing aircrafts landing scenes are characterized by large-scale, no loop closure and plane. Thus, real-time SLAM algorithms [28,29,30] cannot be adopted directly, and a method based on sparse runway features must be developed. Thirdly, in order to operate smoothly under low visibility conditions, a forward-looking infrared (FLIR) camera can be used to monitor the runway, however, a big problem is that fewer features can be extracted from infrared images, especially in the runway area, due to their low resolution and poor texture [31]. Therefore, it is necessary to improve existing algorithms [32,33,34,35] to meet the robustness and accuracy requirements of runway detection in FLIR images. Finally, considering the flight safety, the observability of the proposed visual-inertial navigation system must be analyzed. 

The main contributions presented in this paper are as follows: we propose a visual-inertial landing navigation method based on SR_UKF in which inertial data, infrared images and geo-referenced information are fused to estimate the landing kinetic states of the aircraft. Firstly, a short-wave infrared (SWIR) camera is used to capture a forward-looking infrared (FLIR) image to meet the requirement of precise landing under low visibility. Moreover, the homography that contains the measured pose deviation of the FLIR camera is directly created as the vision observation instead of its decomposition, because it implies the pose deviation between the measured camera and the true camera. Furthermore, an improved runway detection algorithm based on FLIR images is proposed to reach more robustness and accuracy. Specially, a non-linear observability analysis based on Lie derivatives [36,37,38] is performed to ensure that the sensor measurements provide sufficient information for motion estimation. Finally, we design a flight data acquisition platform based on a general aircraft and adopt real flight data to verify that the proposed method can be used for precise landing of commercial aircraft in GPS-denied and low visibility environments. This paper is organized as follows: in Section 2, we propose the visual-inertial navigation system for aircraft precise landing and discuss its observability. Section 3 gives the experimental results and discussions. The conclusions are drawn in Section 4. 

## 2. Methodology

This section explicitly details the framework of the proposed visual-inertial navigation approach. Vision observations with the camera pose deviations are designed elaborately, then the visual-inertial fusion based on SR-UKF is constructed, and its observability is analyzed.

### 2.1. Framework of Infrared-Inertial Landing Navigation

In general, a complete landing procedure of a commercial aircraft includes two parts: an instrument flight segment and a natural vision segment. The instrument portion of an instrument landing procedure ends at the Decision Altitude (DA), and the visual segment begins just below DA and continues to the runway. Prior to reaching DA, the pilot’s primary references for maneuvering the airplane are the aircraft instruments and onboard navigation system. As the pilot approaches the DA, he or she looks for the approach lighting system, if there is one, as well as the runway threshold and touchdown zone lights, markings, surfaces. These visual references help the pilot align the aircraft with the runway and provide position and distance remaining information. At 100 feet above the Threshold Elevation (THRE), the visual transition point, the pilot makes a determination about whether the flight visibility is sufficient to continue the approach and distinctly identify the required visual references using natural vision. If the requirements identified above are met, the pilot may continue descending below DA down to 100 feet height above THRE [39]. Otherwise, the pilot should pull up the aircraft at once, as shown in Figure 1. In order to land commercial airplane safely in GPS-denied and low visibility environments, the pilot needs to obtain accurate navigation information, especially the flight altitude. In the present paper, the proposed method is aimed at landing above 100 feet. 

Among several flight parameters, the flight height is one of the most important ones for the pilot’s decision in a landing procedure. Usually the height measured by barometer or radio altimeter is inaccurate, while the altitude captured by GPS is unreliable. In this paper, the FLIR camera and IMU can complement each other in nature and fuse well in a filtering framework. This paper proposes a novel visual-inertial landing navigation approach based on the SR-UKF, in which visual observation and inertial measurements are integrated to estimate aircraft landing motion. This novel visual-inertial navigation system (VINS) is composed of a FLIR camera, an IMU, a barometer (BARO), a radio altimeter (RALT) and a processing unit that is in charge of motion estimation of the aircraft.

As shown in Figure 2, the inertial measurements are used to propagate the system states, whereas the homography is chosen as the visual observation. The proposed visual-inertial integration can be used for commercial aircraft precise landing in GPS-denied and low visibility.

This method involves three key issues: process modeling, measurement modeling, and its observability. Firstly, this novel vision observation is designed in Section 2.2. Then the visual-inertial navigation based on SR-UKF is proposed in Section 2.3. Finally, the observability of the proposed algorithm is analyzed in Section 2.4. 

### 2.2. Vision Observation

#### 2.2.1. Homography between Synthetic and Real Images

Before proposing the measurement model, we need to analyze the vision measurement mechanism. During the landing, the aircraft descends along the glide slope, and the optical axis of the FLIR camera is aligned with the airport runway. The camera pose is composed of the calibrated IMU/camera relative pose and the measured IMU pose. Ideally the measured camera pose should be equal to the real camera pose. The synthetic image is derived by the terrain data and the measured camera pose, and the real image is captured by the FLIR camera. Therefore, in the image plane the synthetic runway features should be in coincidence with the real detected features accurately. However, the random errors of inertial sensors bring a deviation between the measured camera pose and the real camera pose and further lead to the mismatch between the synthetic runway features and the real runway features. The relationships between the measured camera pose $\left({\hat{\Phi }}^{n},{\hat{P}}^{n}\right)$ and the real camera pose $\left(\Phi^{n},P^{n}\right)$ in the navigation reference frame are described as follows:(1)$$\{\begin{array}{c} \Delta \psi^{n}=\Phi^{n}-{\hat{\Phi }}^{n}\\ \Delta P^{n}=P^{n}-{\hat{P}}^{n} \end{array}$$
where ${\hat{\Phi }}^{n}$ and ${\hat{P}}^{n}$ denote the measured attitude and position of the FLIR camera in the navigation reference frame, respectively, $\Phi^{n}$ and $P^{n}$ represent the real attitude and position of the FLIR camera in the navigation reference frame separately, $\Delta \psi^{n}$ and $\Delta P^{n}$ are the attitude and position measurement deviations of the FLIR camera in the navigation reference frame individually. 

As shown in Figure 3, at the time *t* the transformation from the synthetic image to the real image satisfies the homography $H_{M}^{R}\left(t\right)$, so the synthetic runway features and the real runway features can be understood as two independent visual projections of the same runway from the geodetic coordinate system to the pixel coordinate system, respectively, derived by the real camera pose and the estimated pose. $R_{M}^{R}\left(t\right)$ and $T_{M}^{R}\left(t\right)$ represent the relative rotation and translation of the FLIR camera from the measured pose to the real pose separately. $N_{M}\left(t\right)$ is the unit normal vector of the airport plane with respect to the FLIR camera in the measured pose, $d_{M}\left(t\right)$ denotes the distance from the airport plane to the optical center of the FLIR camera in the measured pose. 

Note that the matrix $H_{M}^{R}\left(t\right)$ depends on the motion parameters {$R_{M}^{R}\left(t\right)$, $T_{M}^{R}\left(t\right)$} as well as the structure parameters {$N_{M}\left(t\right)$, $d_{M}\left(t\right)$} of the ground plane [40,41]. To increase the readability of the mathematical formulae, the time variables in $H_{M}^{R}\left(t\right)$, $R_{M}^{R}\left(t\right)$, $T_{M}^{R}\left(t\right)$, $N_{M}\left(t\right)$ and $d_{M}\left(t\right)$ will be omitted in the sequel. Then, the homography $H_{M}^{R}$ can be expressed as:(2)$$H_{M}^{R}=R_{M}^{R}+\frac{1}{d_{M}}T_{M}^{R}\cdot N_{M}{}^{T}$$

It is notable that the terms $R_{M}^{R}$, $T_{M}^{R}$, $N_{M}$ and $d_{M}$ can be further written in the VINS states as:(3)$$R_{M}^{R}={}^{R}C_{b}^{n}\cdot {\left({}^{M}C_{b}^{n}\right)}^{T}$$
(4)$$T_{M}^{R}={\left({}^{R}C_{b}^{n}\right)}^{T}\cdot \left({}^{M}P_{n}-{}^{R}P_{n}\right)$$
(5)$$N_{M}=-1\cdot {\left({}^{R}C_{b}^{n}\right)}^{T}\cdot e_{3},\;\mathrm{with}\;e_{3}={\left[\begin{array}{ccc} 0 & 0 & 1 \end{array}\right]}^{T}$$
(6)$$d_{M}=-1\cdot e_{3}{}^{T}\cdot {}^{M}P_{n}$$
where ${}^{R}C_{b}^{n}$ is the attitude matrix of the FLIR camera in the real pose, and ${}^{M}C_{b}^{n}$ is the attitude matrix of the FLIR camera in the measured pose. Obviously, the homography matrix contains the deviation between the real camera pose and the measured camera pose, which can be calculated by the line features of synthetic runway and real runway. Furthermore, the synthetic runway features can be derived by geo-information and inertial measurements, and the real runway features can be extracted from FLIR images in real-time. 

#### 2.2.2. Synthetic Runway Features

In the proposed VINS, a FLIR camera and an IMU are installed on the aircraft. As shown in Figure 4, these reference frames obey the right-hand rule in this paper. 

{E} is the Earth-centered earth fixed (ECEF) reference frame, and a point f in {E} is ${}^{E}P_{f}\in {ℜ}^{3}$. {G} denotes the geographic reference frame, any point f in {G} is ${}^{G}P_{f}$. {B} represents the body reference frame. Its origin ${}^{B}O$ is at the center of IMU, $X_{B}$ axis points to the head, $Y_{B}$ axis points toward the right, $Z_{B}$ axis is upward. A point f in {B} denotes ${}^{B}P_{f}\in {ℜ}^{3}$. {C} is the camera reference frame with the origin ${}^{C}O$ at the camera optical center. The $Z_{C}$ axis coincides with the camera principle axis and points to the forward direction. The $X_{C}$ axis points to the column scan direction, while the $Y_{C}$ axis faces to the row scan direction. A point f in {C} is ${}^{C}P_{f}$. {P} denotes the pixel reference frame with its origin ${}^{P}O$ located in the upper-left of image plane. The u− and v− axes in {P} point to the right and downward directions. A point f in {P} denotes ${}^{P}P_{f}\in {ℜ}^{2}$. The runway features in the synthetic image are derived by the runway geographic information and the measured pose of IMU. This vision projection process involves five coordinate transformations as follows:

(1) Transformation between geodetic and Cartesian coordinates in the ECEF reference frame

The geodetic coordinate that contains longitude $L_{i}$, latitude $\lambda_{i}$ and ellipsoidal height $h_{i}$ of any point can be transformed to the Cartesian coordinate in the ECEF reference frame by the following equation: (7)$${}^{E}P_{i}={\left[\begin{array}{ccc} (R_{n}+h_{i})\cdot \cos L_{i}\cdot \cos\lambda_{i}, & (R_{n}+h_{i})\cdot \cos L_{i}\cdot \sin\lambda_{i}, & \left(\left(1-e^{2}\right)\cdot R_{n}+h_{i}\right)\cdot \sin L_{i} \end{array}\right]}^{T}$$
where $R_{n}$ is the radius of curvature in the prime vertical, and e is the first eccentricity of the Earth.

(2) From {E} to {G}

Any known point in the ECEF can be projected into the geographic coordinate system with the IMU center as its origin: (8)$${}^{G}P_{f}=\left[\begin{array}{ccc} -\sin L_{a}\cdot \cos\lambda_{a} & -\sin L_{a}\cdot \sin\lambda_{a} & \cos L_{a}\\ -\sin\lambda_{a} & \cos\lambda_{a} & 0\\ -\cos L_{a}\cdot \cos\lambda_{a} & -\cos L_{a}\cdot \sin\lambda_{a} & -\sin L_{a} \end{array}\right]\cdot \left({}^{E}P_{f}-{}^{E}P_{a}\right)$$
where ${}^{E}P_{f}$ denotes the Cartesian coordinates of any point *f* on the runway surface, ${}^{E}P_{a}$ represents the Cartesian coordinates of the IMU. In order to facilitate the coordinate transformation, the geographic coordinate system {G} is selected as the navigation coordinate system {N}. 

(3) From {N} to {B}

The navigation coordinate system {N} has the same origin with the body coordinate system {B}, the former rotates yaw-pitch-roll angle round $X_{N}-Y_{N}-Z_{N}$ axis to the latter in sequence, as follows:(9)$${}^{B}P_{f}=C_{b}^{n}{}^{T}\cdot {}^{N}P_{f}$$
where $C_{b}^{n}$ denotes the attitude matrix.

(4) From {B} to {C}

The rigid connection between aircraft body and camera contains a relative rotation $R_{B}^{C}$ and translation $T_{B}^{C}$ that has been accurately calibrated before flight: (10)$${}^{C}P_{f}=R_{B}^{C}\cdot {}^{B}P_{f}+T_{B}^{C}$$

(5) From {C} to {P}

According to the pinhole imaging model [42], the homogeneous coordinate projection of any point in the pixel coordinate system is: (11)$${}^{p}P_{f}=\frac{1}{Z_{c}}\left[\begin{array}{ccc} 1/dx & s & u_{0}\\ 0 & 1/dy & v_{0}\\ 0 & 0 & 1 \end{array}\right]\cdot \left[\begin{array}{ccc} f & 0 & 0\\ 0 & f & 0\\ 0 & 0 & 1 \end{array}\right]\cdot {}^{C}P_{f}$$
where *Z_c_* is the normalization coefficient, *dx* and *dy* represent the pixel sizes in image *u* and *v* axes respectively, (*u*_0_, *v*_0_) are the coordinates of the principal point, *s* is the skew parameter, and *f* is the focal length of the FLIR camera. 

Equations (10)–(14) give a complete transformation from the runway plane to the pixel plane of airborne FLIR camera, as shown in Figure 5. Therefore, a marking point ${}^{E}P_{f}$ on the airport runway can be projected onto the pixel plane as a point ${}^{P}P_{f}\in {ℜ}^{2}$. 

Line features of airport runway can be generated by the projection model combining IMU pose and runway geo-information. Consequently, the pixel coordinate of line features can be described as:(12)$$l_{s\to t}={\left[\begin{array}{ccc} 1, & -\left(r_{s}-r_{t}\right)/\left(c_{s}-c_{t}\right), & \left(\left(r_{s}-r_{t}\right)/\left(c_{s}-c_{t}\right)\right)\cdot c_{t}-r_{t} \end{array}\right]}^{T}$$
where ${}^{P}P_{f}={\left[\begin{array}{cc} r_{s} & c_{s} \end{array}\right]}^{T}$ is the pixel coordinate projected from the starting point ${}^{E}P_{s}$, ${}^{P}P_{t}={\left[\begin{array}{cc} r_{t} & c_{t} \end{array}\right]}^{T}$ is the pixel coordinate projected from the terminal point ${}^{E}P_{t}$. 

#### 2.2.3. Real Runway Features

Visible images have high spatial resolution and rich texture details, but these images can be easily influenced by severe conditions, such as poor illumination, fog, and other effects of bad weather. Visible images capture reflected light, whereas infrared images capture thermal radiation. In general, infrared images are resistant to these disturbances. In the present paper, we adopt the SWIR camera to capture FLIR images with important airfield features in low visibility. However, infrared images typically have defects of low resolution and poor texture [31]. Existing runway detection algorithms [32,33,34] cannot satisfy the requirements of robustness and accuracy in airborne extract runway features from FLIR images accurately and robustly. Improvements have been made on the basis of our recently proposed method [35]. In the presented paper, the detection result of the previous image is used as the prior knowledge of the next image to detect and extract four runway edges instead of left and right edges from the FLIR image, as shown in Figure 6. 

This improved method adopts a coarse-to-fine hierarchical idea in which the runway region of interest (ROI) is preliminarily estimated in the FLIR image and the runway edges are finely extracted from the ROI. At the coarse layer, the runway ROI can be calculated by the aircraft pose parameters and airport geo-information in the first few frames. Then, the detected runway is used as the prior knowledge of the next image. Meanwhile, considering the errors of aircraft pose parameters, the runway ROI based on special confidence interval can be estimated. The higher the confidence level is, the larger the runway ROI will be. Therefore, surrounding useless objects and complex background texture can be excluded from ROI so as to reduce interference and image processing time. Especially the errors transfer equations of vision projection model can be given as follows:(13)$$\Delta r=J_{r}\cdot \bar{x},\;\mathrm{with}\bar{x}={\left[\begin{array}{cccccc} \Delta L_{a} & \Delta \lambda_{a} & \Delta h_{a} & \Delta \psi & \Delta \theta & \Delta \phi \end{array}\right]}^{T}$$
(14)$$\Delta c=J_{c}\cdot \bar{x}$$
where Δr is the error of pixel row and Δc is the error of pixel column. $J_{r}$ is the Jacobian of row pixel r with respect to $\bar{x}$, and $J_{c}$ is the Jacobian of column pixel c with respect to $\bar{x}$. $\Delta L_{a}$, $\Delta \lambda_{a}$, $\Delta h_{a}$, Δψ, Δθ, and Δϕ are the measurement deviations of longitude $L_{a}$, latitude $\lambda_{a}$, ellipsoidal height $h_{a}$, yaw ψ, pitch θ, and roll ϕ of the IMU respectively. 

At the fine layer, EDLines detector [43] is used to extract straight line segments from ROI, then fragmented line segments generated by EDLines are linked into complete runway edge lines based on the morphology of synthetic runway in the ROI. Due to the less texture and low resolution of the FLIR image, the detected edges are divided into small segments and scattered in the ROI disorderly. However, each synthetic runway line has a neighborhood which is determined by the pixel errors ( $r-\Delta r\leq \hat{r}\leq r+\Delta r$, $c-\Delta c\leq \hat{c}\leq c+\Delta c$) of its endpoints. If one of the fragmented line segments locates in the neighborhood of any synthetic runway line and the angle between them is less than 3˚, it belongs to the set of the synthetic runway line candidates. Therefore, in the ROI four sets of lines are extracted from the detected line segments individually, and other lines are abandoned. In view of these facts, our method calculates the weight of each line segments according to its length and width. In each set, a number of points are randomly selected from these small line segments according to the line weight value. Obviously, the large weight line segment contributes greatly to the fitting of the line segments. Finally, each set of the line segments can be fitted into an edge line by using the RANSAC method. The detection and extraction results of runway features are given in Section 3.2. 

### 2.3. Visual-Inertial Navigation

The UKF adopts a deterministic sampling technique to estimate the state and covariance of the non-linear models directly. Compared with the EKF, the UKF can predict the state of the non-linear system more accurately rather than calculate the Jacobian and Hessian matrices of the process and measurement models. However, the UKF need calculate the square root of state covariance matrix during sigma points update, it may occasionally generate a negative definite state covariance matrix which will cause the program to abort. The SR-UKF requires less numerical computations and has more accuracy by using a Cholesky factorization of the error covariance matrix in propagation directly [44]. The proposed visual-inertial navigation approach adopts SR-UKF to integrate nonlinear visual observation and inertial measurements to estimate aircraft motion.

#### 2.3.1. Process Modeling

Firstly, we define the system state as: (15)$$x^{T}=\left[\begin{array}{ccccc} \psi^{T} & \delta v^{T} & \delta p^{T} & \varepsilon^{T} & \nabla^{T} \end{array}\right]$$
where $\psi^{n}\in {ℜ}^{3}$, $\delta v^{n}\in {ℜ}^{3}$ and $\delta p^{n}\in {ℜ}^{3}$ are the attitude, velocity and position errors of INS respectively. $\varepsilon^{n}\in {ℜ}^{3}$ denotes the gyroscope drift, $\nabla^{n}\in {ℜ}^{3}$ represents the accelerometer bias. Then the continuous-time system process model is given by: (16)$$\dot{x}\left(t\right)=A\left(t\right)\cdot x\left(t\right)+w\left(t\right)$$
(17)$$A=\left[\begin{array}{ccccc} 0_{3\times 3} & I_{3\times 3} & 0_{3\times 3} & 0_{3\times 3} & 0_{3\times 3}\\ \left[f_{n}\times \right] & 0_{3\times 3} & 0_{3\times 3} & 0_{3\times 3} & C_{b}^{n}\\ 0_{3\times 3} & 0_{3\times 3} & 0_{3\times 3} & -C_{b}^{n} & 0_{3\times 3}\\ 0_{3\times 3} & 0_{3\times 3} & 0_{3\times 3} & 0_{3\times 3} & 0_{3\times 3}\\ 0_{3\times 3} & 0_{3\times 3} & 0_{3\times 3} & 0_{3\times 3} & 0_{3\times 3} \end{array}\right]$$
(18)$$w={\left[\begin{array}{lllll} \varepsilon^{n} & \nabla^{n} & 0_{1\times 3} & w_{g} & w_{a} \end{array}\right]}^{T}$$

Considering the discrete-time, the model can be written as follows:(19)$$x_{k}=\Phi_{k/k-1}x_{k-1}+w_{k-1}$$
(20)$$\Phi_{k/k-1}=e^{\int_{t_{k-1}}^{t_{k}}A(\tau )d\tau }\approx e^{A(t_{k-1})\Delta t}\approx I+A\left(t_{k-1}\right)\Delta t,\;\mathrm{with}\;\Delta t=t_{k}-t_{k-1}$$

#### 2.3.2. Vision Measurement Model

Because the homography matrix contains the deviation of aircraft pose, four groups of possible solutions can be obtained by decomposing the homography matrix according to the traditional method [40,41], and then a set of solutions which are closest to the true value, i.e., the deviation of aircraft pose, can be selected by prior knowledge as UKF measurement. However, the homography matrix decomposition not only increases computation, but also introduces computation errors. In this paper, the measured homography matrix is transformed into one-dimensional column vector, which is used as visual measurement to participate in UKF.

Suppose that ${\hat{H}}_{M}^{R}\in {ℜ}^{3\times 3}$ and $H_{M}^{R}\in {ℜ}^{3\times 3}$ are the measurement and the estimation of the homography, then ${\hat{H}}_{M}^{R}$ and $H_{M}^{R}$ can be converted into two column vectors $vec{\hat{H}}_{M}^{R}\in {ℜ}^{9}$ and $vecH_{M}^{R}\in {ℜ}^{9}$ respectively. Considering the measurement noises of the homography ${\hat{H}}_{M}^{R}\in {ℜ}^{3\times 3}$, the nonlinear vision measurement model is formalized as:(21)$$vec{\hat{H}}_{M}^{R}=vecH_{M}^{R}+v_{flir}$$
where $v_{flir}\in {ℜ}^{9}$ is assumed to be a zero-mean Gaussian noise. 

(1) ${\hat{H}}_{sv}^{ev}$ Calculation

The homography ${\hat{H}}_{M}^{R}\in {ℜ}^{3\times 3}$ can be calculated by the feature matching between synthetic images and real images, which is described in Section 2.3.2. The detailed algorithm for homography calculation refers to [42] which gives the transformation rule for lines. A line transforms as:(22)$$l_{R}={\left({\hat{H}}_{M}^{R}\right)}^{-T}\cdot l_{M}$$
where $\left(l_{R},l_{M}\right)$ is a line pair between the synthetic image and the infrared image. The main line features include the four edges of runway at least that support the calculation of the homography with eight degrees of freedom. 

(2) $H_{M}^{R}$ Estimation

${}^{M}C_{b}^{n}$ is the estimated attitude matrix from the body frame to the navigation frame, and ${}^{M}P_{n}$ is the estimated position of the body, while ${}^{R}C_{b}^{n}$ is the measured attitude matrix, and ${}^{R}P_{n}$ is the measured position of the body. According to Equations (2)–(6), $H_{M}^{R}$ can be calculated as follows:(23)$$H_{M}^{R}={}^{R}C_{b}^{n}\cdot {}^{M}C_{n}^{b}+\frac{{}^{M}C_{n}^{b}\cdot \left({}^{R}P_{n}-{}^{M}P_{n}\right)\cdot e_{3}{}^{T}\cdot {}^{M}C_{b}^{n}}{e_{3}{}^{T}\cdot {}^{M}P_{n}}$$
So $vecH_{M}^{R}$ can be expressed as the function of attitude error $\psi^{n}$ and position error $\delta P^{n}$ through the conversion $H_{M}^{R}\to vecH_{M}^{R}$. 

#### 2.3.3. Other Observations

Besides the above visual measurements, the proposed landing navigation can integrate with other common observations such as air pressure height and radio altitude. These measurement models can be written as follows: (24)$${\hat{h}}_{imu}-{\hat{h}}_{hpr}=\delta h+v_{hpr}=C_{hpr}\cdot \bar{x}+v_{hpr}$$
(25)$${\hat{h}}_{imu}-{\hat{h}}_{ralt}=\delta h+v_{ralt}=C_{ralt}\cdot \bar{x}+v_{ralt}$$
(26)$$C_{hpr}=C_{ralt}={\left[\begin{array}{ccccc} 0_{1\times 3} & 0_{1\times 3} & e_{3}{}^{T} & 0_{1\times 3} & 0_{1\times 3} \end{array}\right]}_{1\times 15}$$
where ${\hat{h}}_{imu}$ is the altitude measured by IMU, ${\hat{h}}_{hpr}$ indicates the air pressure height measured by the barometer, and ${\hat{h}}_{ralt}$ represents the radio altimeter. $v_{hpr}$ and $v_{ralt}$ are all assumed to be zero-mean Gaussian white noise. By combining FLIR vision, air pressure height and radio altimeter, the nonlinear measurement model is presented as:(27)z(t)=C(x)+v(t)
(28)$$z\left(t\right)=\left[\begin{array}{c} vec{\hat{H}}_{M}^{R}\\ {\hat{h}}_{imu}-{\hat{h}}_{hpr}\\ {\hat{h}}_{imu}-{\hat{h}}_{ralt} \end{array}\right],\;C\left(x\right)=\left[\begin{array}{c} vecH_{M}^{R}\\ \delta h\\ \delta h \end{array}\right],v\left(t\right)=\left[\begin{array}{c} v_{flir}\\ v_{hpr}\\ v_{ralt} \end{array}\right]$$

The multi-source information fusion framework based on SR_UKF consists of the process model and measurement model, which realizes the integration of inertial measurements, infrared image, airport geo-reference, air data and radio altitude. 

### 2.4. Observability

Observability is an inherent characteristic of the proposed VINS; it provides an understanding of how well states of a system can be inferred from the system output measurements. Recently there has been many works in studying the observability of VINSs [36,37,38]. We apply the non-linear observability analysis proposed by Herman and Krener in [36] and refer to the work of Kelly [37] and Weiss [38] for details about how to apply this method to a system similar to ours. In the following, the observability analysis of the core system is established by studying the observability matrix rank based on Lie derivatives. 

#### 2.4.1. Nonlinear Observability

Considering the state space as an infinitely smooth manifold X of dimension n, the nonlinear system is described by the following model:(29)$$\{\begin{array}{l} \dot{\chi }=\sum_{i=0}^{p}f_{i}\left(\chi \right)u_{i}\\ y=h\left(\chi \right) \end{array}$$
where $\chi \in {ℜ}^{n}$ is the state vector, $u_{i}\in {ℜ}^{1},i=0\cdots p$ denotes the control input, $u_{0}=1$, and $y={\left[y_{1},\cdots ,y_{m}\right]}^{T}\in {ℜ}^{m}$ is the measurement vector with $y_{k}=h_{k}\left(\chi \right),k=1,\cdots ,m$. The zeroth-order Lie derivative is the function itself, i.e., $L^{0}h\left(\chi \right)=h\left(\chi \right)$. The first-order Lie derivative of h with respect to $f_{i}$ at χ∈X is:(30)$$L_{f_{i}}h\left(\chi \right)=\nabla_{f_{i}}h\left(\chi \right)=\frac{\partial h\left(\chi \right)}{\partial \chi }f_{i}\left(\chi \right)$$

The recursive Lie derivative is defined as:(31)$$L_{f_{j}}L_{f_{i}}h\left(\chi \right)=\frac{\partial L_{f_{i}}h\left(\chi \right)}{\partial \chi }f_{j}\left(\chi \right)$$

The *k*-th derivative of h along $f_{i}$ is:(32)$$L_{f_{i}}^{k}h\left(\chi \right)=\frac{\partial L_{f_{i}}^{k-1}h\left(\chi \right)}{\partial \chi }f_{i}\left(\chi \right)$$

Based on the preceding expression for the Lie derivative, the observability matrix is defined as:(33)$$O=\left[\begin{array}{c} \nabla L^{0}h\left(\chi \right)\\ \nabla L_{f}^{1}h\left(\chi \right)\\ \vdots \\ \begin{array}{c} \nabla L_{f_{i}\cdots f_{j}}^{n}h\left(\chi \right)\\ \vdots \end{array} \end{array}\right]$$

If the observability matrix *O* is full rank, the system is locally weakly observable.

#### 2.4.2. Observability Analysis

In order to reveal the observability of our proposed system, we use the motion state instead of the state errors. The state errors are approximations where second and higher order terms are omitted under the assumption of a small error state [38]. However, the observability analysis on the full nonlinear system prevents information loss. 

First, we define the system state vector of the core system as follows:(34)$$\chi \left(t\right)={\left[\begin{array}{ccccc} q_{b}^{n}{}^{T} & v_{n}{}^{Τ} & p_{n}{}^{T} & b_{g}{}^{T} & b_{a}{}^{T} \end{array}\right]}^{T}$$

Then the nonlinear kinematic equations of the core system for computing the Lie derivatives is rearranged as:(35)$$\left[\begin{array}{c} {\dot{q}}_{b}^{n}\\ {\dot{v}}_{n}\\ {\dot{p}}_{n}\\ {\dot{b}}_{g}\\ {\dot{b}}_{a} \end{array}\right]=\left[\begin{array}{c} -0.5\Xi \left(q_{b}^{n}\right)b_{g}\\ g-C\left(q_{b}^{n}\right)b_{a}\\ v_{n}\\ 0_{3\times 1}\\ 0_{3\times 1} \end{array}\right]+\left[\begin{array}{c} 0.5\Xi \left(q\right)\\ 0_{3\times 3}\\ 0_{3\times 3}\\ 0_{3\times 3}\\ 0_{3\times 3} \end{array}\right]\omega_{m}+\left[\begin{array}{c} 0_{3\times 3}\\ C\left(q\right)\\ 0_{3\times 3}\\ 0_{3\times 3}\\ 0_{3\times 3} \end{array}\right]a_{m}=f_{0}+f_{1}\omega_{m}+f_{2}a_{m}$$
where $C\left(q_{b}^{n}\right)$ is the rotational matrix corresponding to the quaternion $q_{b}^{n}$, Ξ(q) is the quaternion multiplication matrix for the quaternion of rotation q with $\dot{q}=0.5\Xi \left(q\right)\omega$, $\omega_{m}$ denotes the angular velocity vector, $a_{m}$ is the accelerate vector.

A well-known result that we will use in the observability analysis of (31) is the following: when four and more known features are detected in FLIR image frame, the infrared camera pose is observable. According to Equation (2), the measurements can be summarized as:(36)$$h_{1}=C\left(q_{b}^{n}\right)\cdot R_{0}+R_{0}\cdot p_{n}\cdot N_{n}{}^{T}/d_{n}-T_{0}$$
where $R_{0}={}^{\mathrm{sv}}C_{n}^{b}$, $p_{0}={}^{\mathrm{sv}}p_{n}$, $N_{n}=-1\cdot R_{0}\cdot e_{3}$, $d_{n}=-1\cdot e_{3}{}^{T}\cdot p_{0}$, and $T_{0}=R_{0}\cdot p_{0}\cdot N_{n}{}^{T}/d_{n}$. Furthermore, we enforce the unit-quaternion constraints by employing the following additional measurement equation:(37)$$h_{2}={\left(q_{b}^{n}\right)}^{T}\cdot q_{b}^{n}=1$$

(1) Zeroth-Order Lie Derivatives: Define the zeroth-order Lie derivative of $h_{1}$ and $h_{2}$, which are simply the measurement functions themselves, i.e.,:(38)$$L^{0}h_{1}=C\left(q_{b}^{n}\right)\cdot R_{0}+R_{0}\cdot p_{n}\cdot N_{n}{}^{T}/d_{n}-T_{0}$$
(39)$$L^{0}h_{2}={\left(q_{b}^{n}\right)}^{T}\cdot q_{b}^{n}$$

Their gradients are:(40)$$\nabla L^{0}h_{1}=\left[\begin{array}{lllll} \Gamma_{1}\left(q_{b}^{n}\right) & 0_{3\times 3} & D_{1}\left(p_{n}\right)/d_{n} & 0_{3\times 3} & 0_{3\times 3} \end{array}\right]$$
(41)$$\nabla L^{0}h_{2}=\left[\begin{array}{lllll} 2{\left(q_{b}^{n}\right)}^{T} & 0_{3\times 3} & 0_{3\times 3} & 0_{3\times 3} & 0_{3\times 3} \end{array}\right]$$
where $\Gamma_{1}\left(q_{b}^{n}\right)=\frac{\partial C\left(q_{b}^{n}\right)\cdot R_{0}}{\partial q_{b}^{n}}$, $D_{1}\left(p_{n}\right)=\frac{\partial \left(R_{0}\cdot p_{n}\cdot N_{n}{}^{T}\right)}{\partial p_{n}}$. 

(2) First-Order Lie Derivatives: The first-order Lie derivatives of $h_{1}$ and $h_{2}$ with respect to $f_{0}$ are computed as (34):(42)$$L_{f_{0}}^{1}h_{1}=\nabla L^{0}h_{1}f_{0}=-0.5\Gamma_{1}\left(q_{b}^{n}\right)\Xi \left(q_{b}^{n}\right)b_{g}+D_{1}\left(p_{n}\right)v/d_{n}$$
(43)$$L_{f_{0}}^{1}h_{2}=-{\left(q_{b}^{n}\right)}^{T}\Xi \left(q\right)b_{g}$$

Their gradients are:(44)$$\nabla L_{f_{0}}^{1}h_{1}=\left[\begin{array}{lllll} \Gamma_{2}\left(q_{b}^{n}\right) & D_{1}\left(p_{n}\right)/d_{n} & D_{2}\left(p_{n}\right)/d_{n} & -0.5\Gamma_{1}\left(q_{b}^{n}\right)\Xi \left(q_{b}^{n}\right) & 0_{3\times 3} \end{array}\right]$$
(45)$$\nabla L_{f_{0}}^{1}h_{2}=\left[\begin{array}{lllll} \Gamma_{3}\left(q_{b}^{n}\right) & 0_{3\times 3} & 0_{3\times 3} & -{\left(q_{b}^{n}\right)}^{T}\Xi \left(q\right) & 0_{3\times 3} \end{array}\right]$$
where $\Gamma_{2}\left(q_{b}^{n}\right)=\frac{\partial \left(L_{f_{0}}^{1}h_{1}\right)}{\partial q_{b}^{n}}$, $D_{2}\left(p_{n}\right)=\frac{\partial \left(L_{f_{0}}^{1}h_{1}\right)}{\partial p_{n}}$, $\Gamma_{3}\left(q_{b}^{n}\right)=\frac{\partial \left(L_{f_{0}}^{1}h_{2}\right)}{\partial q_{b}^{n}}$.

(3) Second-Order Lie Derivatives: The second-order Lie derivative of $h_{1}$ with respect to $f_{0}$ is computed as (36):(46)$$L_{f_{0}}^{2}h_{1}=\nabla L^{1}h_{1}f_{0}=-0.5\Gamma_{2}\left(q_{b}^{n}\right)\Xi \left(q_{b}^{n}\right)b_{g}+D_{1}\left(p_{n}\right)\left(g-C\left(q_{b}^{n}\right)b_{a}\right)/d_{n}+D_{2}\left(p_{n}\right)v/d_{n}$$

The gradient is:(47)$$\nabla L_{f_{0}}^{2}h_{1}=\left[\begin{array}{lllll} \Gamma_{4}\left(q_{b}^{n}\right) & S\left(v_{n}\right) & D_{3}\left(p_{n}\right) & -0.5\Gamma_{2}\left(q_{b}^{n}\right)\Xi \left(q_{b}^{n}\right) & -D_{1}\left(p_{n}\right)C\left(q_{b}^{n}\right)/d_{n} \end{array}\right]$$
where $\Gamma_{4}\left(q_{b}^{n}\right)=\frac{\partial \left(L_{f_{0}}^{2}h_{1}\right)}{\partial q_{b}^{n}}$, $S\left(v_{n}\right)=\frac{\partial \left(L_{f_{0}}^{2}h_{1}\right)}{\partial v_{n}}$, $D_{3}\left(p_{n}\right)=\frac{\partial \left(L_{f_{0}}^{2}h_{1}\right)}{\partial p_{n}}$.

We obtain the observability matrix *O* by stacking the gradient matrices above: (48)$$O=[\begin{array}{l} \nabla L^{0}h_{1}\\ \nabla L^{0}h_{2}\\ \nabla L_{f_{0}}^{1}h_{1}\\ \nabla L_{f_{0}}^{1}h_{2}\\ \nabla L_{f_{0}}^{2}h_{1} \end{array}]=[\begin{array}{lllll} \Gamma_{1}\left(q_{b}^{n}\right) & 0_{3\times 3} & D_{1}\left(p_{n}\right)/d_{n} & 0_{3\times 3} & 0_{3\times 3}\\ 2{\left(q_{b}^{n}\right)}^{T} & 0_{3\times 3} & 0_{3\times 3} & 0_{3\times 3} & 0_{3\times 3}\\ \Gamma_{2}\left(q_{b}^{n}\right) & D_{1}\left(p_{n}\right)/d_{n} & D_{2}\left(p_{n}\right)/d_{n} & -0.5\Gamma_{1}\left(q_{b}^{n}\right)\Xi \left(q_{b}^{n}\right) & 0_{3\times 3}\\ \Gamma_{3}\left(q_{b}^{n}\right) & 0_{3\times 3} & 0_{3\times 3} & -{\left(q_{b}^{n}\right)}^{T}\Xi \left(q\right) & 0_{3\times 3}\\ \Gamma_{4}\left(q_{b}^{n}\right) & S\left(v_{n}\right) & D_{3}\left(p_{n}\right) & -0.5\Gamma_{2}\left(q_{b}^{n}\right)\Xi \left(q_{b}^{n}\right) & -D_{1}\left(p_{n}\right)C\left(q_{b}^{n}\right)/d_{n} \end{array}]$$
where the complete matrix has size 5 × 5. Considering the system state of aerial vehicle in landing phase, the attitude is relatively stable without any complex maneuver, i.e., pitch θ∈[2°,4°], roll ϕ∈[−1°,1°], and angle velocity vector $\omega_{m}$ is minor. In the observability matrix *O*, these matrices ${\left(q_{b}^{n}\right)}^{T}$, ${\left(q_{b}^{n}\right)}^{T}\Xi \left(q\right)$, and $D_{1}\left(p_{n}\right)$ are full rank. After applying block Gaussian elimination to removing any rows of the matrix *O* that consist entirely of zeros, a row-reduced form of the matrix *O* having the same rank is given by: (49)$$\left[\begin{array}{lllll} 0_{3\times 3} & 0_{3\times 3} & I_{3\times 3} & 0_{3\times 3} & 0_{3\times 3}\\ I_{3\times 3} & 0_{3\times 3} & 0_{3\times 3} & 0_{3\times 3} & 0_{3\times 3}\\ 0_{3\times 3} & I_{3\times 3} & 0_{3\times 3} & 0_{3\times 3} & 0_{3\times 3}\\ 0_{3\times 3} & 0_{3\times 3} & 0_{3\times 3} & I_{3\times 3} & 0_{3\times 3}\\ 0_{3\times 3} & 0_{3\times 3} & 0_{3\times 3} & 0_{3\times 3} & I_{3\times 3} \end{array}\right]$$
which has full column rank, so the proposed system is proven to be observable. 

## 3. Experimental Section and Discussion

In this section, we designed a flight data acquisition platform and adopted real flight data to verify the accuracy and robustness of the proposed method. 

### 3.1. Experiments Preparation

The flight data was gathered at a general aviation airport (Pucheng, China) under different weather conditions such as fog, haze, cloud and sunny. As shown in Figure 7, the general aircraft (Y-12F) was equipped with an image sensing suite, an INS (Applanix AV510), a flight parameter recorder (FPR, AMPEX miniR 700), a flight video recorder (FVR, VM-4), a barometer (BARO, XSC-6E) and a radio altimeter (RALT, Honeywell KRA405b). An image sensing suite (ISS) mounted on the aircraft radome contains a SWIR camera (NIP PHK03M100CSW0) and a visible light camera. Furthermore, an INS, FPR, and FVR were installed on the deck of aircraft cabin. The flight data mainly included FLIR video (frame rate 24 Hz), inertial measurements (update 100 Hz), air pressure height (update 16 Hz) and radio altimeter (update 20 Hz) which were labeled by recorders with time stamp to synchronize measurements. In addition, a DGPS ground station (Trimble R5) was used for DGPS-inertial integration navigation to provide the ground truth.

To get accurate motion estimations, precise FLIR camera parameters and camera/INS relative pose are needed. Classical calibration method based on chessboard pattern [45] is adopted to obtain intrinsic parameters of the FLIR camera. The world coordinates of FLIR camera and INS are individually measured by an electronic total station, then the FLIR camera/INS relative pose can be calculated through vector relation between them [10]. The calibrated parameters of INS and FLIR camera are shown in Table 1.

The flight data is stored in a flight data simulator (FDS), which can play back the whole flight process for the algorithm design and verification. Moreover, the geographic data of the airport and its surrounding has been surveyed accurately. In this paper, experiments are run on an embedded computer board (Nvidia Jetson TX2) with six ARM CPU cores, 256 Pascal GPU cores, 8 GB memory. The block diagram of the experimental platform is shown in Figure 8. The embedded computer receives the airborne sensors data from the FDS and simultaneously reads the airport geographic information stored in the solid-state disk (SSD), then outputs the aircraft motion states through multi-source information fusion. 

### 3.2. Runway Detection Experiment

An ideal line segment detector could process any images regardless of its orientation or size, and extract line segments in real-time without parameters tuning. Among existing algorithms, EDLines detector [19] and Line Segments Detector (LSD) [20] can satisfy these requirements. However, EDLines runs up to 11 times faster than LSD [19], which makes it more suitable for real-time runway detection. As shown in Figure 9, line segments are extracted from the ROI by LSD and EDLines detector, respectively. 

In this paper, a complete landing process in fog is used for verifying the proposed algorithm. Experimental results contain two parts: runway detection and motion estimation. Some runway detection results are shown in Figure 10, From top to bottom the three rows represent three typical scenarios captured at flight altitudes of 200 ft, 100 ft and 60 ft, and the three columns from left to right denote the coarse layer, the fine layer, and the final results, respectively. 

At the coarse layer, the ROI is marked in red at the left column. At the fine layer of our improved method, some line segments in ROI are detected and highlighted in red, and the trapezoid of runway contour is labeled in green at the middle column. These line segments are fitted into the final runway features which are shown in red at the right column. In addition, the statistics of runway detection listed in Table 2 show that the ratio of pixels in ROI to total pixels in CCD is less than 25%. Obviously, the proposed method is faster than others [33,34] which process the whole image, and its robustness is significantly improved. 

### 3.3. Motion Estimation Experiment

As shown in Figure 11, the approach and landing trajectories from the two landing navigation methods are presented. The red curve represents INS/DGPS data, and the green is the motion estimation of the proposed method. The blue pattern denotes the airport runway area. The aircraft descended from 500 feet to 47 feet, through three typical altitudes of 200 feet, 100 feet and 60 feet, flying for 59.45 s. Five recorded time points are marked in this figure. In our experiments, the results of INS/DGPS integration are selected as ground truth.

The proposed algorithm is compared with three other methods such as INS/GPS integration [46], EPnP based method [26] and INS/GPS/BARO/RALT integration [47]. To be consistent with the specifications of the sensor manufactures, the comparison results of position errors, velocity errors, and attitude errors are shown in Figure 12. $\Delta X_{e}$, $\Delta X_{n}$, and $\Delta X_{u}$ denote the measurement errors of the aircraft position in the eastward, northward, and upward, respectively. Δψ, Δθ, and Δϕ represent the measurement errors of the aircraft yaw, pitch, and roll separately. $\Delta V_{e}$, $\Delta V_{n}$, and $\Delta V_{u}$ are the east, north, and azimuth measurement errors of the aircraft velocity severally. As shown in Figure 12, the motion errors of INS/GPS/BARO/RALT integration are obviously larger than those of the others, while the motion errors of the proposed algorithm are smaller than those of the others. Because the EPnP-based algorithm adopts pure image features to calculate the position and orientation of the camera relative to the runway, the accuracy is greatly limited by the relative distance between the camera and the runway. It is difficult to accurately extract the features of the runway terminal in the 500–200 feet stage. Besides, the errors effect of the runway features in the 100–47 feet stage is greater due to the high ratio of runway features to image. The accuracy of motion estimation is higher only in the 200–100 feet stage. 

Meanwhile, the data update rate is limited by the camera frame rate, which is lower than the INS update rate. In addition, the accuracy of motion estimation based on INS/GPS/BARO/RALT cannot be further improved due to the larger measurement errors of barometer and radio altimeter. However, this paper improves the existing runway detection algorithm to avoid the problem that the features of runway terminal are difficult to accurately detect, which can obtain accurate runway features. After the integration of vision measurements and inertial data, the update rate of motion estimation is also improved. Even if in low-visibility environments the motion estimations of the proposed method are still accurate enough, which is benefited from the accurate visual observations. In addition, the RMS errors of different motion estimation are listed in Table 3. The attitude, velocity and position errors of INS/GPS/BARO/RALT integration are slightly larger than those of the others, while these errors of the proposed algorithm are smaller than those of the others.

Among several flight parameters, the height observation is one of the most important for flight safety in landing phase. The flight altitude from GPS is usually inaccurate and unreliable, while the height channel of INS trends to diverges caused by the absence of damping. In general, air pressure height or radio altitude is adopted to damp the height channel of INS, but their accuracy is too low to meet the precision landing. The proposed algorithm that absorbs the advantages of vision and inertial sensors can not only improves the estimation accuracy but also guarantees high update rate.

In Figure 13, the flight height in landing obtained by different methods is represented. The RMS errors of flight height in the landing phase are shown in Table 4. Radio altimeter and barometer are not only of low update rate but also of poor accuracy, which is not suitable for landing navigation.

Although INS/GPS mode has high update rate of height data, its accuracy is poor compared with DGPS/INS. The EPnP-based method has higher accuracy than INS/GPS mode, but EPnP has lower update rate than INS/GPS mode due to its use of pure vision navigation. Obviously, the height precision obtained from the proposed INS/FLIR method is the smallest, it can replace INS/DGPS mode to meet precision landing demands. 

### 3.4. Discussions

The proposed method has high precision up to the DGPS/INS level in low visibility. Firstly, the homography can be served as an ideal visual observation without error accumulation. Meanwhile, owing to the improved runway detection method, it can efficiently overcome the defects of infrared images and smoothly run in a landing scene with large scale and less texture. Compared to ILS and GPS, our method merely takes advantage of an infrared camera to cooperate with airborne navigation sensors, e.g., IMUs, to achieve autonomous motion estimation with low cost, robustness and accuracy. In particular, the accuracy of our method has reached the level of the DGPS/INS for precision approach and landing. 

In the proposed method, the main factors that affect the accuracy of aircraft motion estimation include sensors calibration errors, terrain database precision, spatiotemporal consistency, and runway detection quality. The errors can be partially eliminated by strict sensors calibration [10,45], high precision terrain database and time synchronization [38]. However, the accuracy of runway detection has a great influence on the proposed method, which can be guaranteed by the algorithm itself. The size of synthetic runway neighborhood directly affects the accuracy of the fitted straight line features. If the neighborhood is too small, the line features will not be found. If the neighborhood is too large, the interference features will increase significantly. In this paper, the pixel errors (Δr,Δc) are set to be 2σ, which are trade-off settings.

## 4. Conclusions and Future Works

The paper proposed a novel visual-inertial navigation method to provide drift-free pose estimation for fixed-wing aircrafts landing, in which inertial measurements, infrared observations and geo-information are organically fused in the UKF. In addition, the proposed method has been proven to be observable by nonlinear observability. Comprehensive experiments with real flight data have verified the accuracy and robustness of the proposed method. 

In the future, there are still some research tasks to do for further improvement. (1) For stronger adaptability, we will adopt a multispectral image fusion method [48,49] to enhance the sensitivity in more weather conditions such as rain, snow, or dust. (2) Deep-learning methods [50] can be tried to detect semantic objects with known geo-references around the runway in infrared images, which should not only increase the quantity of vision features to improve the system precision, but also intensify the robustness to detect and recognize different airports. (3) For convenience, the online technique for calibration of the camera to an inertial system [51,52] can also be used to substitute the complicated hand-eye calibration.

## Figures and Tables

**Figure 1 sensors-19-00408-f001:**
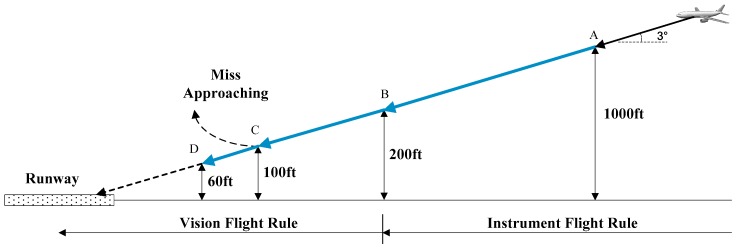
Approach and Landing procedure.

**Figure 2 sensors-19-00408-f002:**
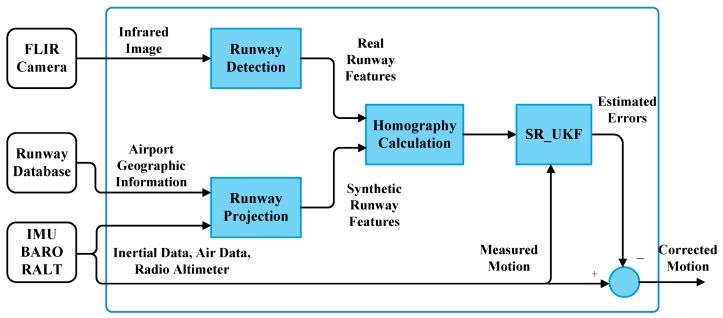
Framework of the proposed landing navigation: the blue box is the core part of the proposed approach

**Figure 3 sensors-19-00408-f003:**
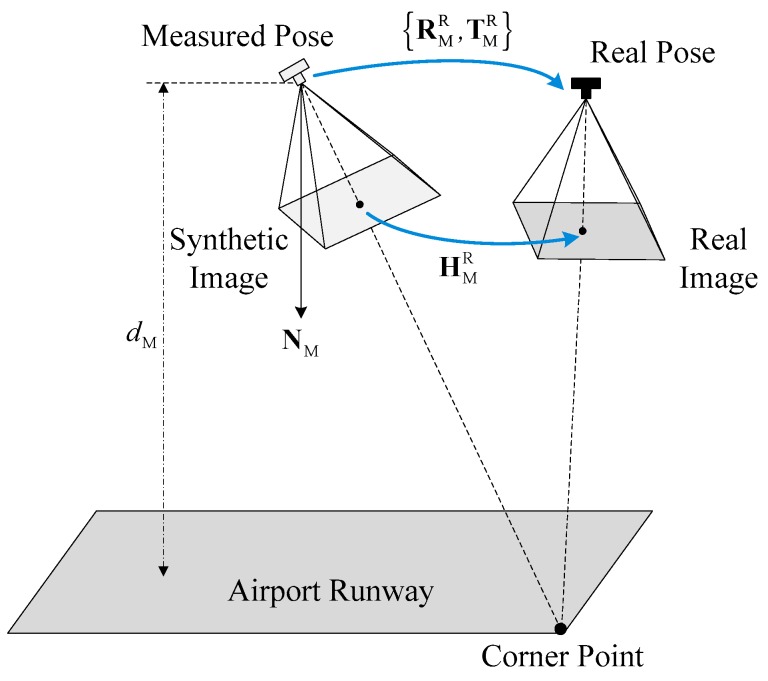
Homography between synthetic and real images

**Figure 4 sensors-19-00408-f004:**
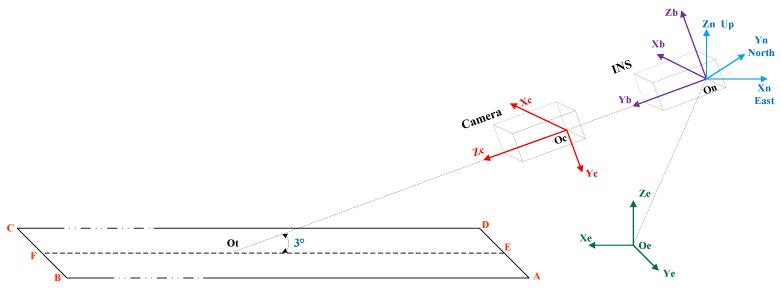
Reference frames and runway model

**Figure 5 sensors-19-00408-f005:**
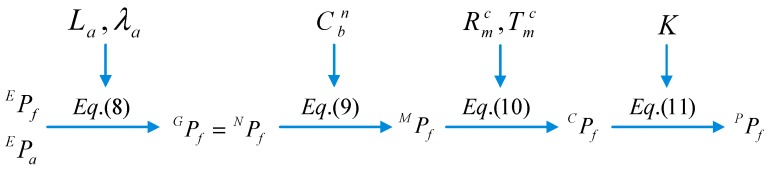
Projection of features in synthetic image.

**Figure 6 sensors-19-00408-f006:**
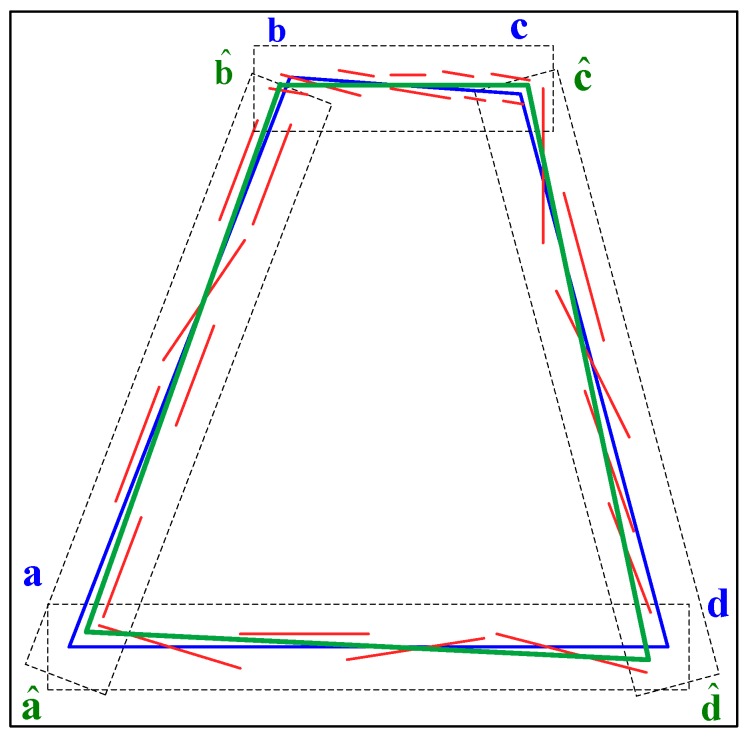
Real Runway Detection: the black solid rectangle is the runway ROI, the red lines are the extracted line segments, the blue quadrangle is the synthetic runway contour, the black dashed rectangles are the neighborhoods of runway edges, and the green quadrangle is the fitted runway edge.

**Figure 7 sensors-19-00408-f007:**
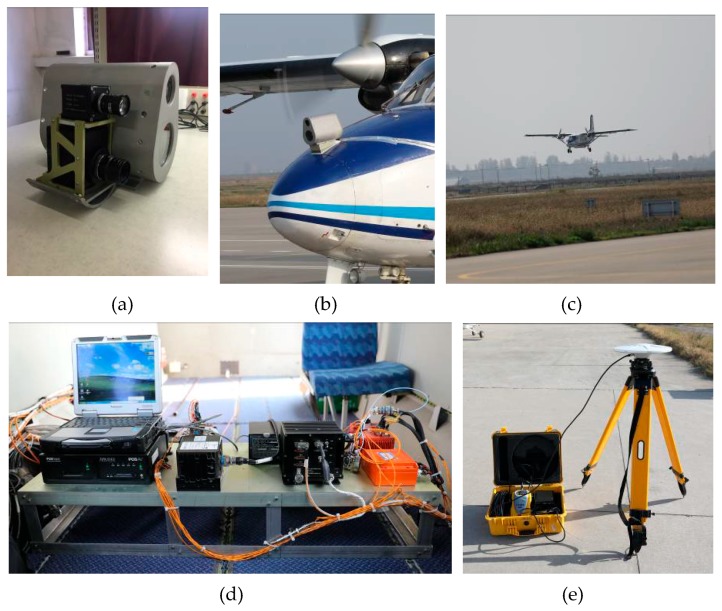
The flight data acquisition platform: (**a**) ISS; (**b**) ISS installation; (**c**) aircraft landing; (**d**) instruments for flight data acquisition; (**e**) DGPS ground station.

**Figure 8 sensors-19-00408-f008:**
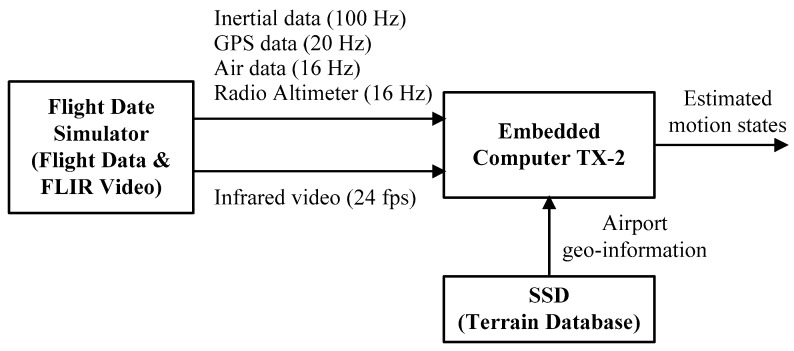
The block diagram of the experimental platform.

**Figure 9 sensors-19-00408-f009:**
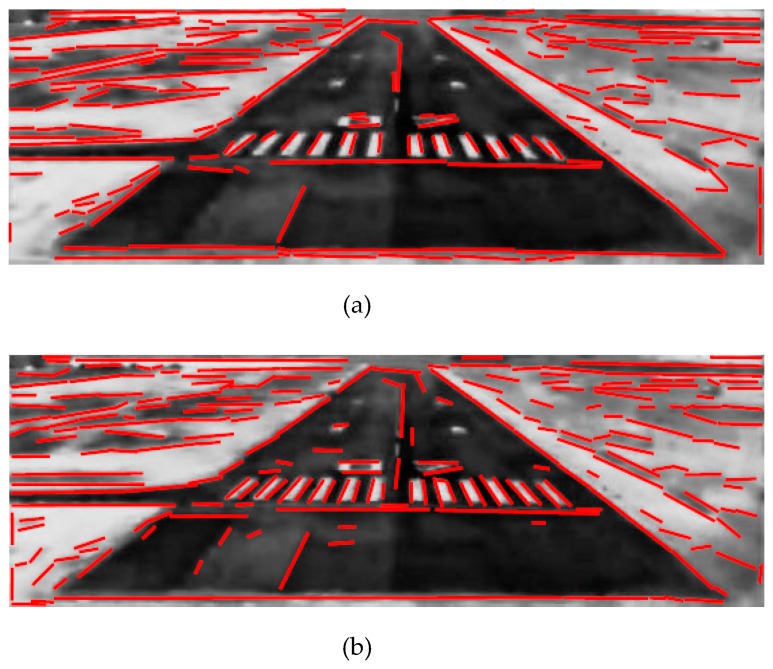
Line Segments Extraction from ROI: (**a**) EDLines: 173 lines, 3.1 ms; (**b**) LSD: 213 lines, 17.1 ms.

**Figure 10 sensors-19-00408-f010:**
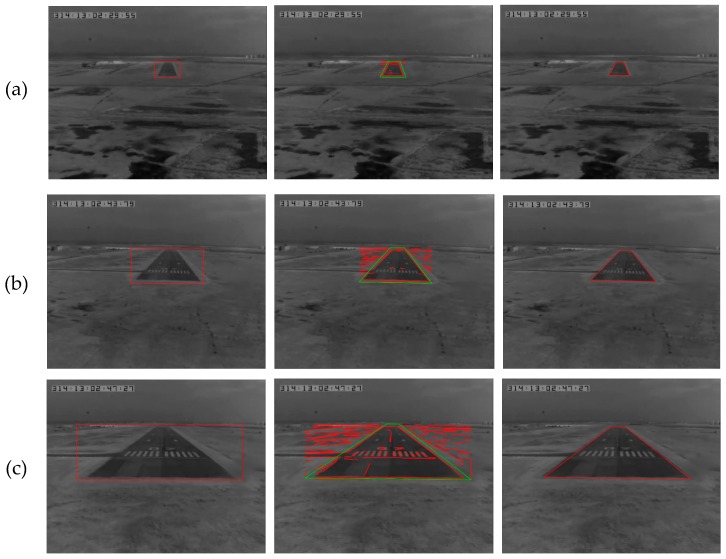
Runway detection at typical flight height: (**a**) 200 ft; (**b**) 100 ft; (**c**) 60 ft.

**Figure 11 sensors-19-00408-f011:**
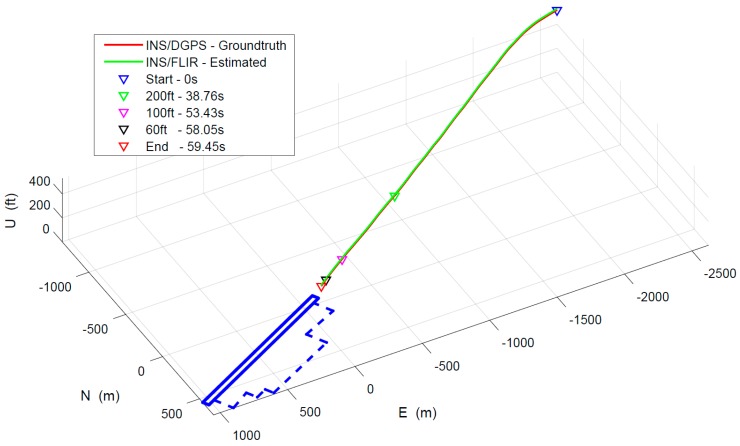
Approach and landing trajectory.

**Figure 12 sensors-19-00408-f012:**
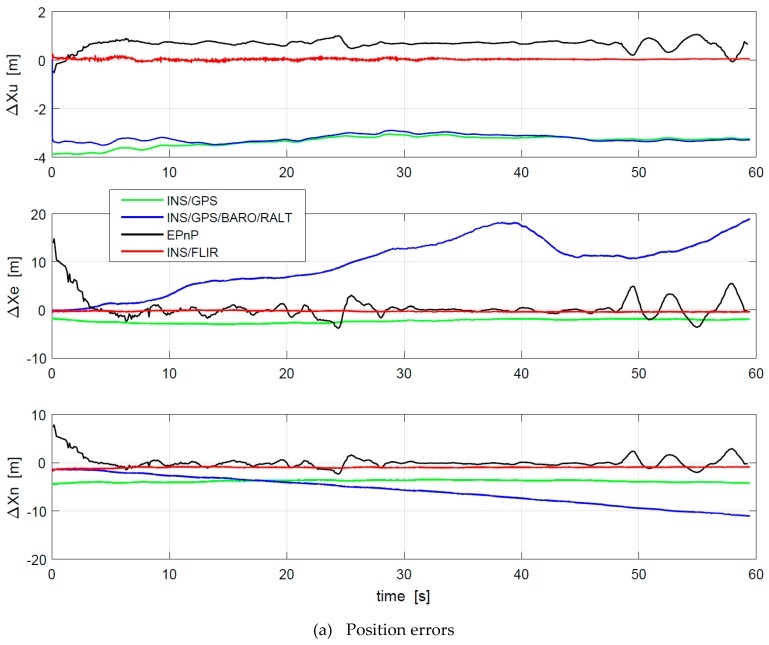
Errors of motion estimation: (**a**) position errors, (**b**) attitude errors, and (**c**) velocity errors.

**Figure 13 sensors-19-00408-f013:**
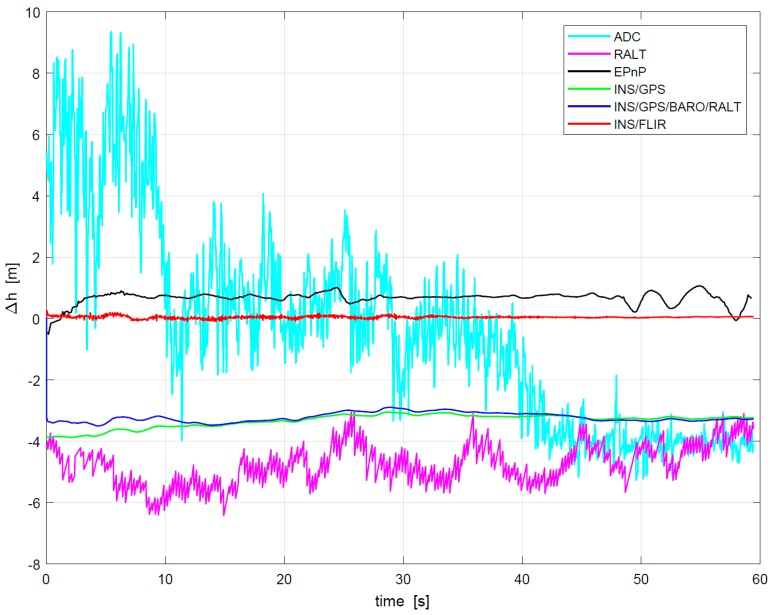
Flight height among 6 modes during landing.

**Table 1 sensors-19-00408-t001:** The calibrated parameters of INS and FLIR camera.

FLIR Camera Intrinsic Parameters	pixel size	0.025 µm
	focal length	*f_x_* = 1010.7 pixel, *f_y_* = 1009.5 pixel
	principal point	*u*_0_ = 316.376 pixel, *v*_0_ = 237.038 pixel
	radial distortion	*k*1 = −0.3408, *k*2 = 0.1238
	spectral response	0.9–1.7 μm
	CCD resolution	640 × 512
	field of view	20°(H) × 30°(V)
FLIR Camera Installation	position	[−0.002, 0.094, −12.217] m [−0.0181, −0.0824, −0.0049] rad
	attitude	
INS Installation	position	[0.0704, −0.4742, −7.2863] m [0.0789, 0.0003, −0.0088] rad
	attitude	

**Table 2 sensors-19-00408-t002:** The statistics of the runway detection at three typical flight altitude.

Scenarios	Flight Height (Ft)	ROI (pixels)	ROI/CCD Ratio	Lines
1	200	49 × 77	0.0115	16
2	100	106 × 214	0.0692	58
3	60	164 × 488	0.2442	173

**Table 3 sensors-19-00408-t003:** RMS Errors of attitude, velocity and position.

Height	Method	Δθ (deg)	Δ*ϕ* (deg)	Δ*ψ* (deg)	Δ*Ve* (m/s)	Δ*Vn* (m/s)	Δ*Vu* (m/s)	Δ*Xe* (m)	Δ*Xn* (m)	Δ*Xu* (m)
500–200 ft	A	0.0222	0.0275	0.0275	0.1384	0.2344	0.1180	0.2544	1.0091	0.0638
	B	0.5252	0.0118	0.0478	0.1300	0.1205	0.1118	2.5186	3.8128	3.3936
	C	0.3686	0.0345	0.0808	0.0980	0.6693	0.1159	9.3157	4.3762	3.2143
	D	0.2056	0.1645	0.0118	—	—	—	2.4090	1.3071	0.6973
200–100 ft	A	0.0133	0.0275	0.0151	0.1096	0.1709	0.0748	0.3938	0.6297	0. 4013
	B	0.5063	0.0151	0.0303	0.0916	0.1164	0.0754	1.9192	3.7881	3.2277
	C	0.4934	0.0364	0.0650	0.1001	0.4617	0.0780	12.952	8.6355	3.2394
	D	0.4415	0.3349	0.0207	—	—	—	1.5185	0.7632	0.6902
100–60 ft	A	0.0122	0.0268	0.0203	0.0793	0.1228	0.0601	0.4051	0.8981	0.0531
	B	0.4869	0.0135	0.0080	0.0802	0.1093	0.0619	2.0229	4.0380	3.2476
	C	0.4773	0.0346	0.0375	0.1173	0.3777	0.0631	14.617	10.401	3.3094
	D	0.6914	0.5275	0.0304	—	—	—	2.5647	1.3917	0.7617
60–47 ft	A	0.0190	0.0379	0.0131	0.1187	0.1282	0.1189	0.4038	0.8795	0.0567
	B	0.4762	0.0596	0.0141	0.0769	0.1525	0.1221	1.9263	4.1950	3.2373
	C	0.4703	0.0800	0.0486	0.1390	0.5439	0.1224	18.051	10.948	3.2825
	D	0.7734	0.4161	0.0202	—	—	—	3.3010	1.7121	0.4384

Note: A—INS/FLIR, B—INS/GPS, C—INS/GPS/BARO/RALT, D—EPnP.

**Table 4 sensors-19-00408-t004:** RMS Errors of flight height in landing phase, unit (m).

Height	INS/FLIR	INS/GPS	INS/GPS/BARO/RALT	EPnP	BARO	Radio Altimeter
500–200 ft	0.0638	3.3936	3.2143	0.6973	3.0746	4.9590
200–100 ft	0.0413	3.2277	3.2394	0.6902	3.6841	4.7333
100–60 ft	0.0531	3.2476	3.3094	0.7617	4.0300	4.1154
60–47 ft	0.0567	3.2373	3.2825	0.4384	4.1483	3.6150

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
