# Peer review of "Infrared-Inertial Navigation for Commercial Aircraft Precision Landing in Low Visibility and GPS-Denied Environments"

_sensors, 2019, doi:10.3390/s19020408_

Reviewer 1 Report

The paper is a very interesting example of how inertial measurements and infrared images acquired from an aircraft coupled with airport geo-information, can be combined by means of a filtering method in order to aid landing in poor visibility conditions and when the onboard radio-navigation systems are ineffective.

The paper is particularly well written (only a few misspelling) and the cited references (52 entries) are abundant and relevant. The methodology (section 2) is generally clearly explained, particularly the Square-Root Unscented Kalman Filter (SR-UKF) involved in the data processing. I enjoyed reading the observability analysis presented in section 2.4.2, that theoretically  substantiates the method of data fusion. The experimental results presented in section 3 are rather convincing.

I would like, however, to make some suggestions in order to clarify some mathematical aspects of the paper. These suggestions and some minor corrections are directly written down in the printed version of the paper in attachment. A document that describes the WGS84 geodetic reference system have been annexed to the attached file. This document includes the relationships which have to be used to transform geodetic coordinates into Cartesian coordinates.

Author Response

Dear Reviewer,

    We would like to express our heartfelt gratitude to you for taking the time to read our manuscript. We would like to sincerely thank for your valuable comments to help improve the paper. Please find our responses in the Word file.

Thank you and best regards.
Yours sincerely

Reviewer 2 Report

The paper presents a new methodology for instrument precision approach and landing. The method is based on swir vision system fused with ins data using a square-root unscented Kalman filter. The vision data is obtained by homography decomposition and compared with synthetic image deduced from geo-database inferred from camera pose corrected by ins data system. The paper is very well written and free of errors, except for a minor correction in line 204 regarding the Zb axis orientation. The proposed system is analytically shown to be observable and the theoretical results are corroborated by a very detailed experiment with a sophisticated flight instrumentation system. The present paper definitely a good contribution to the safe operation of commercial aircraft in the near future.

Author Response

Dear Reviewer,

We would like to express our heartfelt gratitude to you for taking the time to read our manuscript. We would like to sincerely thank for your valuable comments to help improve the paper. Please find our responses in the enclosed file.

Thank you and best regards.
Yours sincerely

Reviewer 3 Report

The authors proposed a fusion scheme between IMU, IR camera and synthetic 3D world (composed of the runway) for estimating the full state of a plane during landing phase. Other sensors are added during the estimation phase such as a barometer and a radio altimeter.

Each part of the framework is not innovative by itself, as IMU/Vision fusion already exists, visual servoing based on synthetic data is very common and UKF filtering is classical in such estimation problem. Considering the vision part, the relative positioning is based on a very simple Homography estimation and the main problem of the approach (the detection of the landmarks) is treated in a previous work with existing methods (EDLines and LSD). The observability study has also come from state-of-the-art works and applies to this system to prove the viability of the solution.

The main novelty is the airplane landing application.

Considering the introduction, the state-of-the-art is very clear and complete even if sometime some deeper analysis would be welcome to well understand the difference between the proposed approach and the other works.

The theoretical part is well described and may allow readers to do the same work by themselves.

Considering the experimental part, all the results are presented very clearly and are easily understandable. The improvement given by the proposed method seems also very good.

To conclude, this article is very clear and interesting as it presents a complete framework for a given interesting application. Even if the novelty is limited, I think that this paper is very interesting for the community as it provides an easy framework to tackle an important problem.

Author Response

(The authors gave the same response as above.)
